

# Technical note: Retrieval of the supercooled liquid fraction in mixed-phase clouds from Himawari-8 observations

Ziming Wang[1,2], Husi Letu[3], Huazhe Shang[3], Luca Bugliaro[1]

[1]Institute of Atmospheric Physics, Deutsches Zentrum für Luft- und Raumfahrt (DLR), Oberpfaffenhofen, 82234, Germany
[2]Meteorological Institute, Ludwig Maximilian University of Munich, Munich, 80333, Germany
[3]State Key Laboratory of Remote Sensing Science, Aerospace Information Research Institute, Chinese Academy of Sciences, Beijing, 100101, China

*Correspondence to*: Ziming Wang (Ziming.Wang@dlr.de)

**Abstract.** The supercooled liquid fraction (SLF) in mixed-phase clouds (MPCs) is an essential variable of cloud microphysical
processes and climate sensitivity. However, SLF is currently calculated in spaceborne remote sensing only as the cloud phase frequency ratio of adjacent pixels, which results in a loss of the original resolution in observations of cloud liquid or ice content within MPCs. Here, we present a novel method for retrieving the SLF in MPCs based on the differences in radiative properties of supercooled liquid droplets and ice particles at visible and shortwave-infrared channels of the geostationary Himawari-8. Liquid and ice water paths are inferred by assuming that clouds are composed of only liquid or ice, with the real cloud water
path (CWP) expressed as a combination of these two water paths (SLF and 1-SLF as coefficients), and SLF is determined by referring to the CWP from CALIPSO. The statistical relatively small cloud phase spatial inhomogeneity in Himawari-8-pixel level indicates an optimal scene for the cloud retrieval. The SLF results are comparable to global SLF distributions observed by active instruments, particularly for single-layered cloud systems. While accessing the method's feasibility, SLF averages are estimated between 74% and 78% in Southern Ocean stratocumulus across seasons, contrasting with a range of 29% to 32%
in Northeast Asia. The former exhibits a minimum SLF around midday in summer and a maximum in winter, while the latter trend differs. This novel algorithm will be valuable for research to track the evolution of MPCs and constrain the related climate impact.

## 1 Introduction

Clouds cover about 70% of the earth's surface and significantly impact the hydrologic cycle and radiative budget (Stephens et
al., 2012; Watanabe et al., 2018). Liquid water or ice form in clouds and fall back to Earth as precipitation. By reflecting incident solar radiation and trapping upwelling radiation within the atmosphere, clouds result in an imbalance of radiation budget. Mixed-phase clouds (MPCs), composed of both supercooled liquid droplets and ice particles at temperatures between 0 and -38 °C (Pruppacher and Klett, 1997), are recognized as the great sources of uncertainty in precipitation formation and cloud radiative properties (McCoy et al., 2014; Mülmenstädt et al., 2015; Tan et al., 2016). Cloud ice particles form and grow
at the expense of supercooled liquid droplets in MPCs (Bergeron, 1935), and these microphysical processes govern MPCs'



lifecycle and precipitation formation (IPCC, 2021). Moreover, Lohmann (2002) and Sassen and Khvorostyanov (2007) show that the net radiative impact of MPCs decreases as supercooled liquid droplets glaciate. Regarding the cloud climate feedback, if the supercooled liquid water is underestimated in MPCs, the overestimated ice water in a warmer climate will melt to more cloud water with the higher reflection and let more energy back to the space, and then offset global warming (Bjordal et al.,

2020) about 2 °C at most (Tan et al., 2016). Therefore, a comprehensive investigation of the supercooled liquid fraction (SLF) in MPCs is imperative to reduce uncertainties related to precipitation (Silber et al., 2021) and cloud-climate feedback (Gettelman and Sherwood, 2016; Lohmann and Neubauer, 2018).

Presently, numerous researchers have focused on assessing the SLF and detangling the associated precipitation and climate impact (Mülmenstädt et al., 2015; Cesana et al., 2017; Henneberger et al., 2023). Laboratory experiments, for instance, DeMott

(1990), measure the fraction of soot acting as ice nucleating particles (INPs) for cloud ice, but cannot guarantee a good representation of the real atmospheric conditions. Ramelli et al. (2021) correlate the liquid, mixed-phase, and ice cloud regimes with in situ measured SLF, and calculate the ice particle multiplication with respect to ice nucleating particles – the opposite direction of the SLF, but field experiments are naturally limited due to the small sample volume and rarely repeated samplings in the same cloud. Besides, different global climate models (GCMs) are generally unable to simulate the variation in the SLF

with temperature as obtained from satellite data (Komurcu et al., 2014; Desai et al., 2023).

To obtain a wider range of spatiotemporal scales, polar orbiting satellite observations are exploited to investigate the SLF. The experiments using the CERES-CloudSat-CALIPSO-MODIS satellite dataset (Kato et al., 2011) and radiative transfer calculations show that the cloud tops usually have larger SLF over the Southern Ocean (SO) (Bodas-Salcedo et al., 2016), where the energy budget is poorly simulated in the absorbed shortwave radiation due to the role of cloud (Trenberth et al.,

2010; Huang et al., 2021). Additionally, Choi et al. (2010), Tan et al. (2014), Zhang et al., (2015), Li et al., (2017), Kawamoto et al. (2020), and Villanueva et al. (2020, 2021) find that the variation of the SLF is generally negatively correlated with the occurrence of dust aerosols - effective INPs - using either the Cloud-Aerosol Lidar and Infrared Pathfinder Satellite Observations (CALIPSO) feature mask, or the combined observations from active remote sensing (CALIPSO and CloudSat) and passive remote sensing (Moderate Resolution Imaging Spectroradiometer  MODIS and Polarization and Directionality of

Earth Reflectances POLDER). This finding is obvious in the northern hemisphere (NH) due to a large amount of anthropogenic emissions. Once the aerosol effect on nucleation is not predominant, Li et al., (2017) also demonstrated the associated relationship between SLF and meteorological conditions in Northeast Asia. In these studies, however, only phase frequency ratio, the ratio of liquid pixels to the total liquid and ice cloudy pixels in adjacent pixels of satellite images, is calculated. This treatment loses the original resolution of observations of cloud liquid water or ice content in MPCs, and cannot investigate the

change of the SLF throughout the life span of clouds easily.

Geostationary passive satellite observations not only enable to capture the evolution of clouds at a larger scale (Letu et al., 2019; Wang et al., 2019) other than a narrow swath observed from polar orbiting satellites, but also hold facility to retrieve the cloud liquid water or ice mass content (Nakajima and Nakajima, 1995; Kawamoto et al., 2001; Platnick et al., 2017; Heidinger et al., 2020; Stengel et al., 2020). Coopman et al. (2019) have simulated radiances of MPCs with a radiative transfer model





(RTM) and concluded that the significant variation of cloud effective radius (CER) can serve to depict the cloud phase transition from liquid to ice (the change of SLF) on a cloud tracking algorithm. With these insights, we design an experiment to investigate the potential to retrieve the ratio of liquid content to total liquid and ice content, SLF, in MPCs using the new-generation geostationary satellite Himawari-8 observations, based on the cloud microphysics retrieval. Cloud liquid water or ice content of a certain path, in another word, cloud water paths (CWP), are calculated using cloud optical thickness (COT)

and CER retrieved based on the traditional cloud retrieval scheme (Nakajima and King, 1990) with satellite radiances at visible (VIS) and shortwave infrared wavelength (SWI) channels. The used ice particle shapes or liquid droplets are determined in advance, as their radiative properties may result in differences in the derived microphysical properties (Baum et al., 2014; Holz et al., 2016; Letu et al., 2016; Yang et al., 2018). Thus, differences from radiative properties of ice particles and supercooled liquid droplets due to their refractive indices, sizes, concentration and shapes can probably be used to test the possibility to

determine the cloud ice or water mass (Sun and Shine, 1994), and subsequently ice fraction (IF) and the SLF in MPCs. Nagao and Suzuki (2021, 2022) developed a temperature-independent cloud phase retrieval method that relies on differences between observed and simulated radiances under the assumptions of either liquid or ice water when retrieving COT and CER. This approach consolidates our concept of the SLF retrieval.

In this paper we introduce an algorithm to retrieve the SLF by exploiting different radiative properties of liquid droplets and

ice particles at VIS and SWI wavelengths, and its first attempts on the investigations of the diurnal cycle of SLF in distinct cloud regimes across seasons and hemispheres. To the authors' knowledge, this is the first quantitative analysis of the SLF in the single pixel level of passive remote sensing, especially for the geostationary satellite observations with the typically broadest spatiotemporal scales. MPCs, which we focus on, are an important cloud phase with occurrences around 27% (Mayer et al., 2023) over the field of the geostationary Meteosat satellite, and the SLF within this cloud phase brings about the primary

source of uncertainty in estimating climate sensitivity. The paper is organized as follows: the collocated datasets, the simulated radiative properties of liquid droplets and ice particles, and the concrete SLF retrieval procedure are described in Sect. 2; the main results, including the first retrieval of the SLF in MPCs for the selected cases, the statistical cloud phase spatial inhomogeneity, the validation with the CALIPSO-GOCCP dataset, and the feasibility of the method to investigate the diurnal cycle of SLF in different cloud systems are discussed in Sect. 3; and the conclusion is given in Sect. 4.

## 2 Data and Methodology

### 2.1 Data collocation and preprocessing

The SLF retrieval method uses Himawari-8 spectral data as the main input. The Advanced Himawari Imager (AHI) imager on board Himawari-8 has 3 VIS, 3 SWI, and 10 longwave infrared (LWI) channels (listed in Table 1) with a spatial resolution of 5 km (to be consistent with the used official cloud phase product) and a time interval of 10 min for full disk observations (60°S

- 60°N and 80°E - 160°W; Bessho et al., 2016). Each cloudy pixel is classified into (1) ice, (2) water, and (3) MPCs, using reflectance ratios and brightness temperature differences (Mouri et al., 2016). Our retrieval is capable of providing the SLF in



MPCs from this Himawari-8 official product. The MPCs here consist of various conditions: liquid on top of ice MPC or vice versa, or liquid and ice in the same cloud layer in one pixel. Cloud top temperature (CTT) is retrieved based on channel observations, vertical profiles and cloud type data and used to define the temperature isotherm for SLF. The reflectance at VIS

and SWI is effective for retrieving cloud microphysical properties. The CALIPSO cloud water path $CWP_{CALIPSO}$ (Garnier et al., 2021) estimated using CALIPSO Imaging Infrared Radiometer (IIR) effective emissivity datasets and the microphysical index (Parol et al., 1991) provides the truth value of cloud water and ice. The CALIPSO orbit track passes over the Himawari-8 disk between approximately 3:00 and 7:00 UTC each day. We collocate the Himawari-8 spectral data with CALIPSO tracks for January, May, August, and October in 2017, covering regions such as the North China Plain, the Tibetan Plateau, the Indian

and Pacific Ocean and the SO. The time difference is constrained to be the closest, and the resolution of $CWP_{CALIPSO}$ is interpolated from 333 m to 5 km. AHI pixels that do not have similar cloud top heights (CTHs) across all CALIPSO pixels are removed. In total, the number of collocated observations for model training and validation is 336685 samples. To build the CWP prediction model for the full disk of Himawari-8, AHI channel and geometrics data (solar zenith angle SZA, viewing zenith angle VZA, and Relative Azimuth Angle RAA), Aqua MODIS 16-day averaged surface albedo (resolution: 0.05°, grid

to the AHI resolution using the nearest neighbor algorithm), and $CWP_{CALIPSO}$ serve as the input and the output, respectively (see Table 1).

**Table 1: Overview of the variables used in the CWP model building. Native resolution of Himawari-8 channel datasets and CALIPSO IIR observations: 5 km and 333 m.**

|  | Source | Variables | Range of values | Unit |
|---|---|---|---|---|
|  |  | VIS band | 0.46, 0.51, 0.64 | µm |
|  |  | SWI band | 0.86, 1.6, 2.3 | µm |
| Input | Himawari-8 Level 1 & Level 2 | LWI band | 3.9, 6.2, 7.0, 7.3, 8.6, 9.6, 10.4, 11.2, 12.3, 13.3 | µm |
|  |  | SZA | $0 \sim 90$ | ° |
|  |  | VZA | $0 \sim 90$ | ° |
|  |  | RAA | $0 \sim 180$ | ° |
|  |  | Cloud phase | mixed phase | - |
| Output | MODIS 16-Day CALIPSO Level 2 IIR | Surface albedo | 0.1-1 |  |
|  |  | Ice Liquid Water Path | $1 \sim 1300$ | g/m² |

To investigate the potential uncertainty of the SLF retrieval from the cloud phase vertical and horizontal inhomogeneity, we use the "ice" and "water" flags found within the "Feature_Classification_Flags" parameter of CALIPSO Level 2 Vertical Feature Mask (VFM) product (spatial resolution: 333m, Hu et al., 2009) during the same period in 2017. Here the cloud phase is distinguished by the depolarization ratio and layer-integrated backscatter intensity measurements, with correlation



coefficients exceeding 0.5 in over 90% of warm water clouds (Hu et al., 2009). The single-layered cloud fraction is defined as
the number of single-layered cloud profiles divided by the number of total cloud sample profiles at every 5° of latitudes for
the analysis of zonal distribution. Here, we not only consider the topmost cloudy pixel in the VFM product but also include a
vertical extension of 180 m (6 and 3 boxes at altitudes of 0-8.2 and 8.2-20.2 km) starting from cloud top.
We categorize cloudy boxes as belonging to distinct cloud layers if there is a vertical cloud gap of at least 2 km between them.
The horizontal scale of liquid and ice phase in the single-layered cloud system is determined by multiplying the number of
continuous cloud top phase profiles and the 333 m along-track resolution of CALIPSO.

The lidar-only CALIPSO-GOCCP climatology (Chepfer et al., 2010) developed to facilitate the evaluation of GCMs is
exploited for the validation of the SLF retrieval. This product is derived from the original CALIPSO lidar measurements, and
contains observational cloud diagnostics consistent with the "lidar simulator" (resolution: 2 ° in latitude and longitude, Chepfer
et al., 2008). The "Ratio Ice / (Ice + Liq)", the relative percentage of ice in cloud with respect to the total condensate, in the
"3D_CloudFraction_phase" dataset of CALIPSO-GOCCP is considered as the "true" IF. Himawari-8 retrievals for MPCs are
averaged every 2 ° between 60 °S to 60 °N to collocate the nearest CALIPSO-GOCCP grids and ensured with the closest
match between two groups of CTHs.

## 2.2 Radiative properties of liquid droplets and ice particles

The foundational theory behind SLF retrieval lies in the differences in radiative properties between liquid droplets and ice
particles. Before delving into their optical properties, we have an analysis of cloud phase as a function of CTT observed by
Himawari-8, considering the direct correlation between temperature and cloud glaciation. As depicted in Supplement S1, the
distribution of ice phase, MPCs (including supercooled), and liquid clouds aligns with expected temperature ranges - water
phase dominates at CTT larger than 273 K, ice phase occurs at CTT smaller than 233 K, and MPCs are between these
thresholds.
Subsequently, we start to choose appropriate ice particle habits, considering their distinct scattering capabilities. In a global
scale, the relatively high occurrence rate of supercooled water is consistent with that of droxtals, which both appears at a
latitude band of about 30° at high-levels in both the NH and Southern Hemispheres (SH) (Sato and Okamoto, 2023). Yang et
al. (2003) also demonstrate that droxtals may provide a realistic representation of the smallest nonspherical ice crystals. In the
ice cloud habit model employed in MODIS Collection 5, droxtals are also assigned a 100% habit fraction for particle diameters
below 60μm (Baum et al., 2005). Thus, we incorporated droxtals, spherical crystals, and Mie scattering databases for ice
particles and liquid droplets into the RSTAR7 RTM (Nakajima and Tanaka, 1986), and analyze the sensitivity of radiative
properties to thermodynamic phase states and particle shape. The radiative transfer simulations were performed using the U.S.
standard atmospheric profile. With this RTM, the Himawari-8 radiance in the 2.3 μm channel (SWI 2.3), which is used to
retrieve the official cloud microphysical product of Himawari-8 because of its sensitivity to both COT and CER for thin clouds,
is simulated.





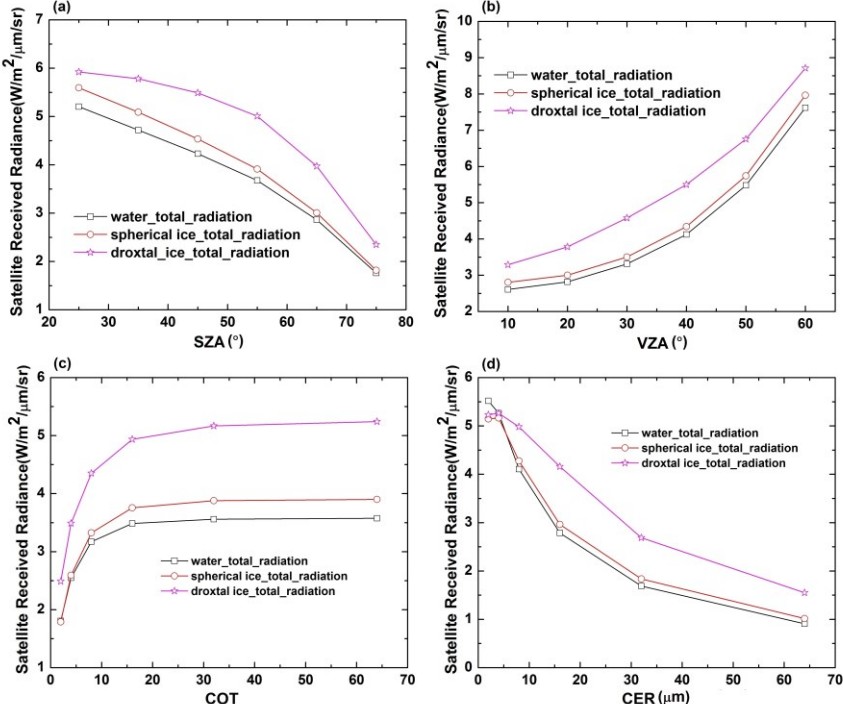

**Figure 1: Sensitivity of satellite radiance at 2.3 µm channel for liquid droplets, spherical ice particles, and droxtals to different parameters. a. VZA=30°, RAA=180°, COT=10, CER=12µm. b. SZA=60°, RAA=180°, COT=10, CER=12µm. c. SZA=60°, VZA=30°, RAA=180°, CER=12 µm. d. SZA=60°, VZA=30°, RAA=180°, COT=10. For all cases surface albedo=0.1, cloud bottom and top height 3~5 km.**

In Fig. 1 we present the sensitivity of the SWI 2.3 radiance to liquid droplets or ice particle shapes with respect to SZA, VZA, COT, and CER. In Fig. 1a, radiance decreases with increasing SZA, while in Fig. 1b an increasing trend with VZA is shown. Fig. 1c reveals satellite-observed radiance generally rising as COT increases, stabilizing beyond COT=15. Conversely, in Fig. 1d, it decreases with rising CER, with the rate of decrease gradually slowing. The results confirm that changes in COT and CER have a great impact on radiance. With respect to the influence from selected droplets or ice particles, the radiance is generally highest for droxtals and lowest for liquid droplets. The disparity due to thermodynamic phase between spherical ice particles and liquid droplets is not significant, whereas the simulated radiance from droxtals is much larger than that of spherical ice particles, which confirms the importance of ice particle shape assumptions in the RTM.





## 2.3 Retrieval procedure

COT and CER provide the basics for the calculation of CWP. The differences between the liquid water path $LWP_{SCs}$ and ice water path $IWP$ of assumed SCs and ice clouds, composed of liquid droplets or droxtals only, and the reference $CWP_{CALIPSO}$, can be used to evaluate the SLF in MPCs.

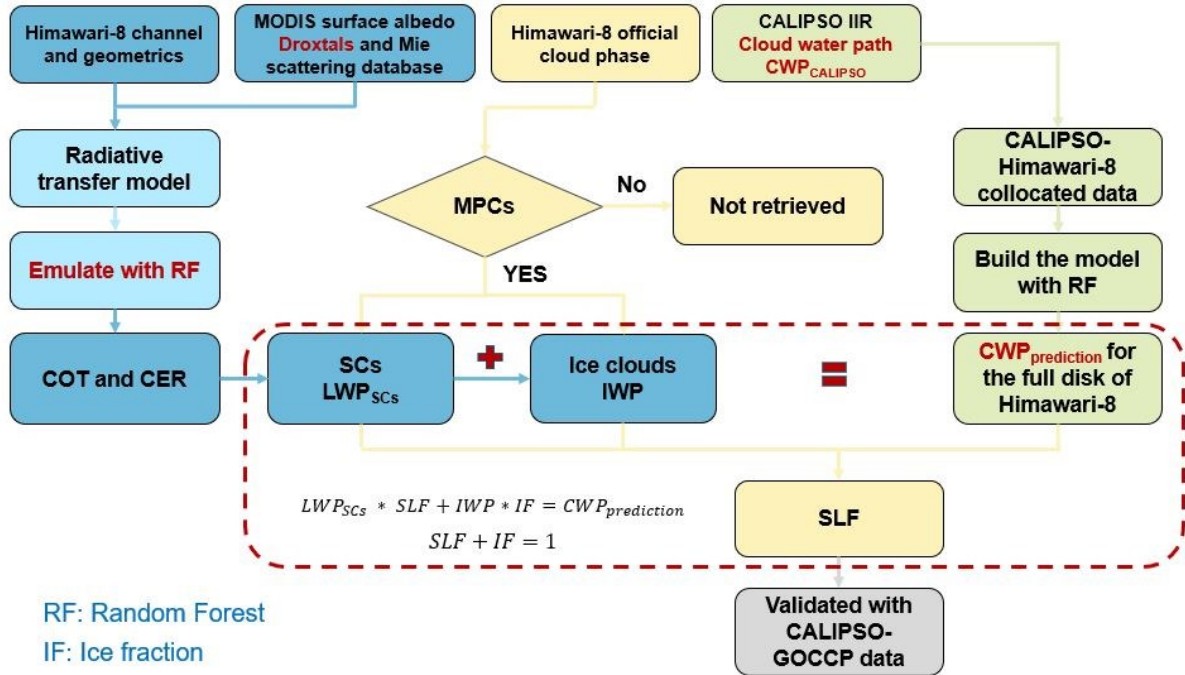

**Figure 2: Roadmap of the SLF retrieval process and subsequent validation. The retrieval process is delineated into three distinct**
**steps, each indicated by color-coded rectangles: blue, green, and yellow.**

    The SLF retrieval for MPCs consists of 3 main steps and the corresponding procedure is shown in Fig. 2. (1) We retrieve CWP under two extreme assumptions: a) the cloud is fully liquid (supercooled) and ice does not contribute to the spectral observations; b) the cloud is fully glaciated and liquid droplets do not affect the satellite radiances. This provides us with $LWP_{SCs}$ and $IWP$. It is conducted by means of the conventional cloud retrieval scheme by Nakajima and King (1990) that
COT and CER can be determined by the radiances at VIS and SWI. The scheme is applied with RSTAR7 RTM calculations. To speed up the technique, the Random Forest (RF) technique (Ri et al., 2022), widely used in regression problems by randomly combining multiple decision trees and demonstrating good performance even in the presence of many unknown features and noise within the dataset, is used to emulate the RTM calculations for VIS 0.64 and SWI 2.3 available from Himawari-8. And the COT and CER are derived when the RTM emulator radiance pairs are closest to the actual satellite
observations (for details, see Text S2, Fig. S2 and Table S1 in the supplement). Then $LWP_{SCs}$ and $IWP$ can be calculated based on the retrieved COT and CER using the following Eq. (1) described in Stephens (1978),

$$CWP = \frac{4COT \times CER \times \rho}{3Q_e},$$ (1)



where ρ is the water or ice density of 1000 kg/m³ and 917 kg/m³, $Q_e$ is the mean extinction coefficient of water or ice.

(2) In addition to $LWP_{SCs}$ and $IWP$ we derive the total $CWP_{prediction}$ for the MPCs pixels. To this end, we develop another

RF model that is trained with mainly channel data (see Table 1) from Himawari-8 observations and the "Ice_Liquid_Water_Path" from the CALIPSO IIR dataset $CWP_{CALIPSO}$ as input and output, respectively, based on the CALIPSO-Himawari-8 collocated measurements. Similar with the model training for the RTM, we set "n_estimators", "criterion", and "min_samples_leaf" in RF as 100, mean square error, and 1. The collocated data set was split into training (90%) and test data (10%) randomly. The CWP prediction model demonstrates high accuracy, achieving an R-squared (R2)

value of 0.97 and mean absolute error (MAE) of 8.32 g/m² between the model simulations and the individual test data. With this procedure, the reference $CWP_{prediction}$ can be predicted for the full Himawari-8 disk. (3) For every MPCs pixel in the official Himawari-8 product, we assume that $CWP_{prediction}$ is the weighted mean of $LWP_{SCs}$ and $IWP$, two extreme situations where no ice or no liquid water is contained in the cloud.

$$LWP_{SCs} \times SLF + IWP \times IF = CWP_{prediction} \text{,} \tag{2}$$

$$SLF + IF = 1 \text{,} \tag{3}$$

The two coefficients SLF and IF in the combination in Eq. (2) represent the relative contribution of liquid and ice to the total CWP and satisfy the condition that SLF=0 when IF=1 (only ice) and vice versa for purely supercooled clouds. Thus, we set the sum of SLF and IF to one in Eq. (3), and Eq. (2) can be solved for SLF. This way, SLF is retrieved for the first time from a cloud water and ice mass equation for every single pixel.

In the detected MPCs, liquid droplets and ice particles may be uniformly distributed or spatially separated, and the horizontal lengths scale vary from 100 km down to 100 m according to airborne measurements (Korolev and Milbrandt, 2022). The ideal scenario for a cloud retrieval is that liquid droplets or ice particles group into large separate pockets and reduce the inhomogeneity. For that reason, the cloud phase spatial inhomogeneity is estimated (Sect. 3.2). In the cloud retrieval scheme, the bi-spectral reflectance method with VIS and SWI (e.g., at 1.6, 2.3 or 3.7 µm) is exploited. The shorter wavelength shows

the larger sensitivity to the CER at a deeper location within the cloud, with the SWI 2.3 in this study being susceptible to CER below the cloud tops. In addition to the cloud phase horizontal extension, the presence of cloud phase overlap scene poses the difficulty for algorithm application. Moreover, the variety of ice particle habits arising from multiple particle growth regimes (Huang et al., 2021) renders the assumption of using a singular ice particle shape - droxtals - simplistic. Thus, larger uncertainty and possible invalid SLF retrieval are expected for ice-dominant MPCs, such as some warm conveyor belt (WCB) or deep

tropical convections. Negative values of SLF are set to zero for these invalid cases, and they are consequently excluded from the statistical analysis. The uncertainties in the developed method may arise due to the limited dataset. So as to verify if the combination coefficients correspond to the real SLF and IF, the retrieved SLF is validated with the CALIPSO-GOCCP product on the global scale (Sect. 3.3), and in the vertical direction from lidar measurements for different cloud regimes to verify the feasibility of the method (Sect. 3.4).



## 215  3 Results and Discussions

### 3.1 Retrieval of the SLF in mixed-phase clouds

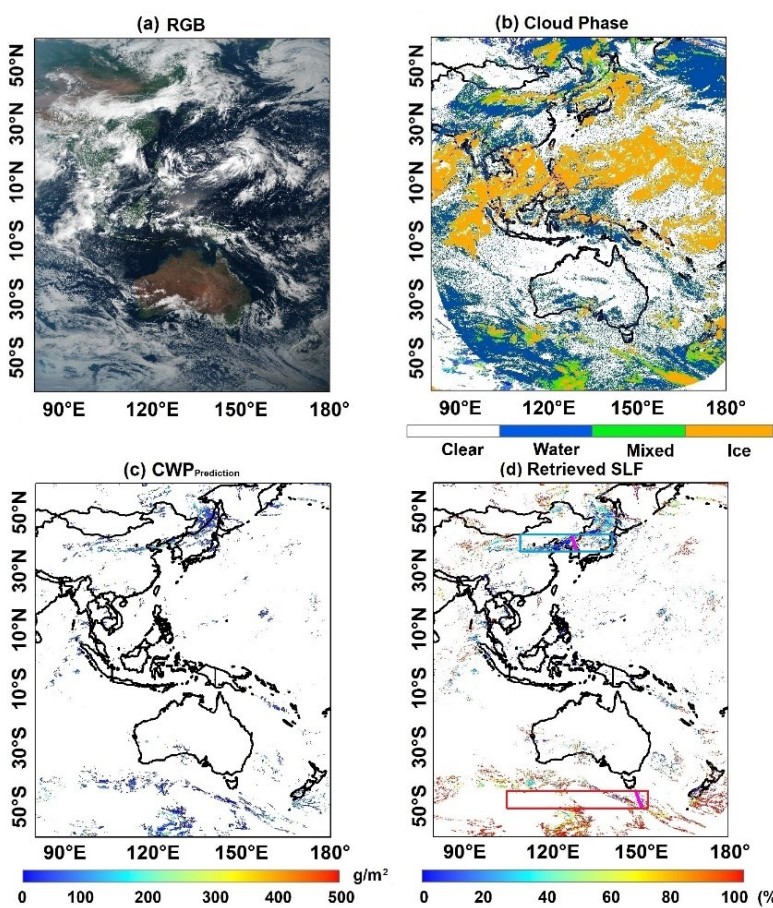

**Figure 3: The general cloud situation and retrieved SLF in the detected MPCs at 4:00 UTC on 28 August 2017. a. RGB. b. Cloud top phase. c. $CWP_{prediction}$ based on the collocated Himawari-8 and CALIPSO observations. d. SLF in MPCs. In panel d, the**
**magenta short lines, as well as the red and blue rectangles, depict the cloud systems alongside CALIOP orbit tracks, and illustrate the diurnal cycle of SLF within two cloud regimes to be examined in Sect. 3.4.**

As a first application, we perform our retrieval on satellite observations on 28 August 2017 at 4:00 UTC and the outputs are

presented in Fig. 3 for the scene of MPCs from the official Himawari-8 product. In Fig. 3a, the RGB image, and Fig. 3b, the

cloud phase classification, we show that MPCs are more prevalent over the middle latitudes, with a particular concentration

observed over the SO and Northeast Asia. In agreement with the RTM simulations in Fig. 1, the COT and CER retrieved for

the assumed fully supercooled clouds in Fig. S3a and S3c are slightly larger and obviously smaller than those of the pure ice

clouds in Fig. S3b and S3d (For details please refer to the Fig. S3 in the supplement). The $CWP_{prediction}$ in Fig. 3c derived

based on CALIPSO-Himawari-8 collocated measurements are expected to lie between the values of $LWP_{SCs}$ and $IWP$ from

the combination of Fig. S3a, c and Fig. S3b, d. For this case, the SLF increases from the subtropics to upper middle latitudes,



and SLF larger than 80% is frequently observed over the SO, related to either the deactivation of INPs from the increasing sulfate aerosol coagulation (Hu et al., 2010), or the low availability of INPs due to the remoteness of anthropogenic emissions (Matus and L'Ecuyer, 2017). The occurrence of larger SLF is less frequent in areas where ice cloud tops prevail, and notably, a region with the "smallest SLF value belt" (< 30%) is evident in Northeast Asia.

## 3.2 The cloud phase spatial inhomogeneity

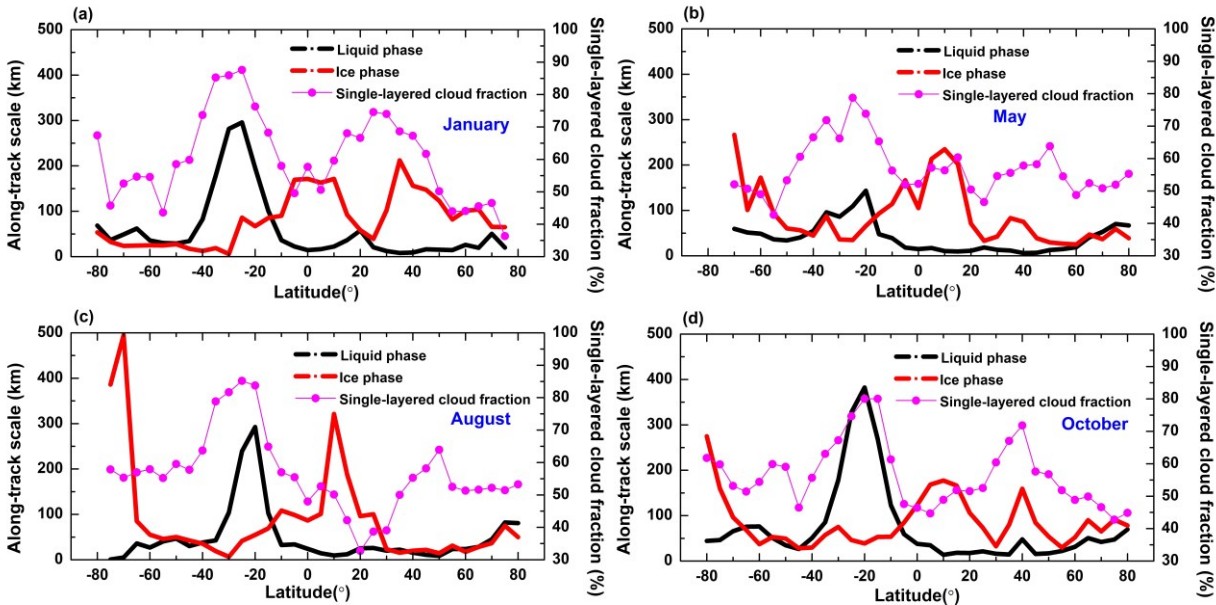


**Figure 4: The zonal distribution of averaged single-layered cloud fractions, and the associated cloud phase along-track horizontal scales from the CALIPSO VFM product in (a) January, (b) May, (c) August, and (d) October of 2017.**

As the passive satellite cloud retrieval is based on the typical single-layered cloud assumption, biases are introduced under the conditions of multilayered cloud systems (Cloud layers interlace at a certain distance from each other) (Li et al., 2015). To

depict three-dimensional structures of clouds, we collect the CALIPSO VFM datasets and plot the zonal distributions of the seasonal averaged single-layered cloud fractions, together with cloud liquid and ice phase along-track horizontal scales (see Fig. 4). From our statistical results we show that the seasonal variations in single-layered cloud percentages are small. The high-value and low-value centers of the fractions (magenta lines) stand out. One major minimum lower than 45% occurs in the tropics in Fig. 4d, and two minor minima occur in the middle to high latitudes; two local peaks up to 87% occur in the

subtropics to middle latitudes in Fig. 4a, from major stratocumulus-dominated oceanic areas.

The along-track horizontal scales of cloud liquid or ice in single-layered cloud systems have obvious zonal and seasonal variations. The liquid phase (black line) has minimum scales (approximately 10-15 km) in the tropics and poleward of 40°N/S, and maximum scales (up to 382 km) at 20°S in Fig. 4d. The ice phase generally has the opposite distribution of horizontal scales, with a common maximum larger than 320 km at subtropics, which can be caused by dissipating convection and the



subsequent horizontal cirrus anvils. The local maximum during spring (Fig. 4b, red line) in the northern midlatitudes may be due to the influences of high-level dust on ice nucleation (Choi et al., 2010; Tan et al., 2014; Villanueva et al., 2020). In general, cloud phase tends to cluster as large pockets (Korolev and Milbrandt, 2022), and the cloud phase spatial inhomogeneity is scarcely shown in the Himawari-8-pixel level (5km).

### 3.3 Comparison with the CALIPSO-GOCCP

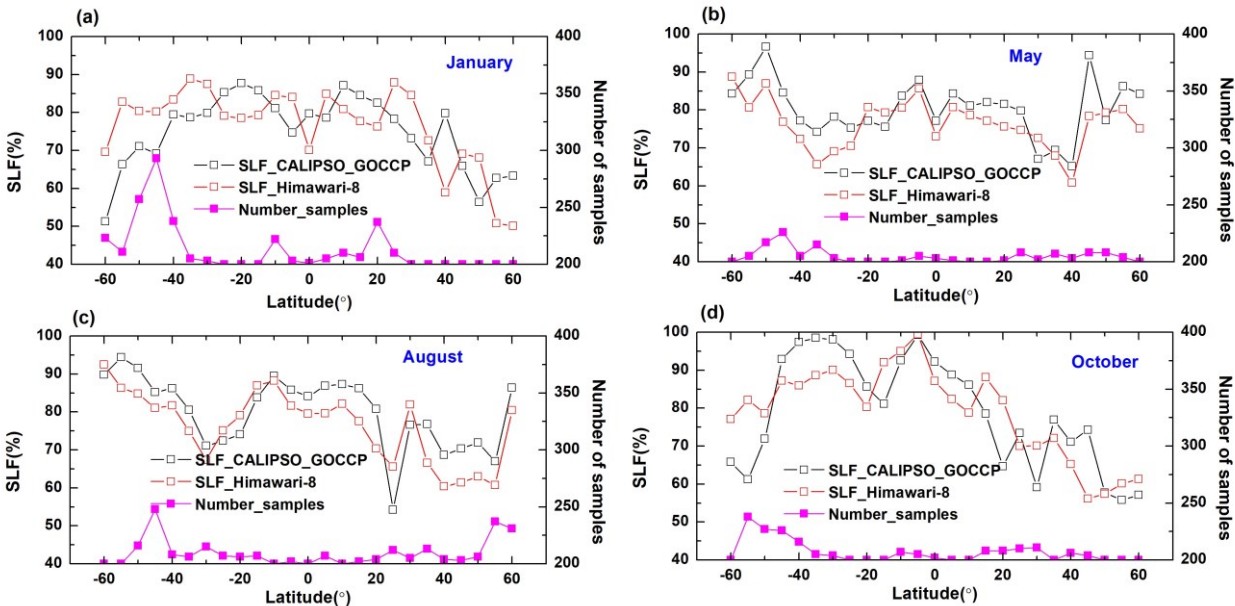

**Figure 5: The zonal distribution of averaged SLF from Himawari-8 retrieved results and the CALIPSO-GOCCP in (a) January, (b) May, (c) August, and (d) October of 2017.**

To further estimate the accuracy of our SLF retrieval, we perform an evaluation using CALIPSO-GOCCP, which is independent of IIR $CWP_{CALIPSO}$ for training the model in Sect. 2.3. In Fig. 5 we present the comparison between Himawari-8

retrieval results and CALIPSO-GOCCP, focusing on hemispherical and seasonal differences in the mean SLF. The magenta lines represent the number of collocated MPCs pixels as a function of latitude (longitude: 80°E - 160°W). Similar to the SLF pattern in the case study discussed in Sect. 3.1, low SLF values are detected from approximately 20° to 40° latitude, particularly over the NH. This is depicted in Fig. 5c, where the monthly average value hits a minimum of 54%. This phenomenon can be attributed to the inverse relationship between SLF and aerosol loadings (Choi et al., 2010; Hu et al., 2010; Tan et al., 2014; Li

et al., 2017; Villanueva et al., 2020, 2021). The zonal mean SLF values exhibit notably higher levels at higher latitudes, particularly evident in May in the NH and during both May and August in the SH. The reason is that the correlation between SLF and surface temperature is almost negative with decreasing temperature (Li et al., 2017). The MPCs over the SO during austral autumn (Fig. 5b) and winter (Fig. 5c) show the larger proportion of supercooled liquid, reaching approximately 90%. The SLF in MPCs around tropical regions is generally shown as a "Valley" but not that low with the SLF higher than 70%,



despite the dominance of ice clouds in this area. The smaller SLF is possibly due to strong precipitation exhausting the large supercooled liquid droplets. The numbers are consistent with the statistical results of the zonal mean cloud SLF for medium clouds (3.36 - 6.72 km) from Fig. 2c in Guo et al., (2020).

Table 2 records the validation of the SLF retrieval for 2017. The average MAE, root mean square error (RMSE) and correlation coefficient (CC) are 9.18%, 10.76%, and 0.69, respectively. Figure 5 illustrates that the accuracy of the SLF retrieval decreases

during austral autumn and winter, notably in January and October in the NH, primarily attributed to the misclassification of snow-covered surfaces as ice clouds on the complex terrain, from the high surface visible reflection detected by Himawari-8. In May and August, the MAE can decrease to as low as 7.38%, and the CC can reach as high as 0.80. In contrast, in January and October, the highest RMSE is 13.25%, the MAE can increase to 10.24%, and the CC can decline to 0.60. Overall, the SLF retrieval is comparable to the CALIPSO-GOCCP lidar observations. The results in Fig. 5 indicate that retrieval accuracy is at

its peak around 20°S, where single-layered cloud systems prevail, as demonstrated in Fig. 4. The differences between the SLF retrieval and CALIPSO-GOCCP are more apparent in low SLF regimes but diminish when SLF values are higher. The SLF retrieval method can further help investigate the global distribution of cloud glaciation.

**Table 2: The validation of retrieved SLF in MPCs with the CALIPSO-GOCCP in 2017.**

| MAE (%) | MAE (%) | RMSE (%) | CC |
|---------|---------|----------|-----|
| January | 10.24 | 13.25 | 0.62 |
| May | 7.38 | 8.03 | 0.72 |
| August | 7.54 | 8.67 | 0.80 |
| October | 11.56 | 13.10 | 0.60 |
| All | 9.18 | 10.76 | 0.69 |

**3.4 The feasibility of the retrieval in investigating the diurnal cycle of SLF across cloud regimes**

To verify the effectiveness of the new algorithm application in different cloud regimes, the collocated lidar products are reused for validating the performances of the SLF retrieval in MPCs from the Himawari-8 official cloud phase products in selected geographic locations. Given the limited size of the dataset, our focus here is on discussing the feasibility of the developed retrieval method.

For example, Fig. 6 shows the CALIPSO-GOCCP product overlaid with the Himawari-8 SLF retrieval along the CALIPSO overpass over the SO and the WCB in the NH over land on August 28, 2018 (two magenta lines in Fig. 3d). In Fig. 6a, an altostratus-stratocumulus overlap structure is observed over the ocean, characterized by an SLF consistently exceeding 85%, except for the MPCs dominated by ice crystals located around 48°15' S. Conversely, in the WCB depicted in Fig. 6b, the SLF consistently remains below 50%. Notably, the retrieved SLF shows better agreement with the CALIOP-GOCCP over the

oceanic areas, and the deviation is always within the range of ±10%. The underestimation of the retrieved SLF can be found



in the WCB, particularly around 38° N with the bias up to 20%. This finding is consistent with the discussion on the algorithm uncertainty from cloud phase overlap and the assumption of ice particle shapes in Sect. 2.3.

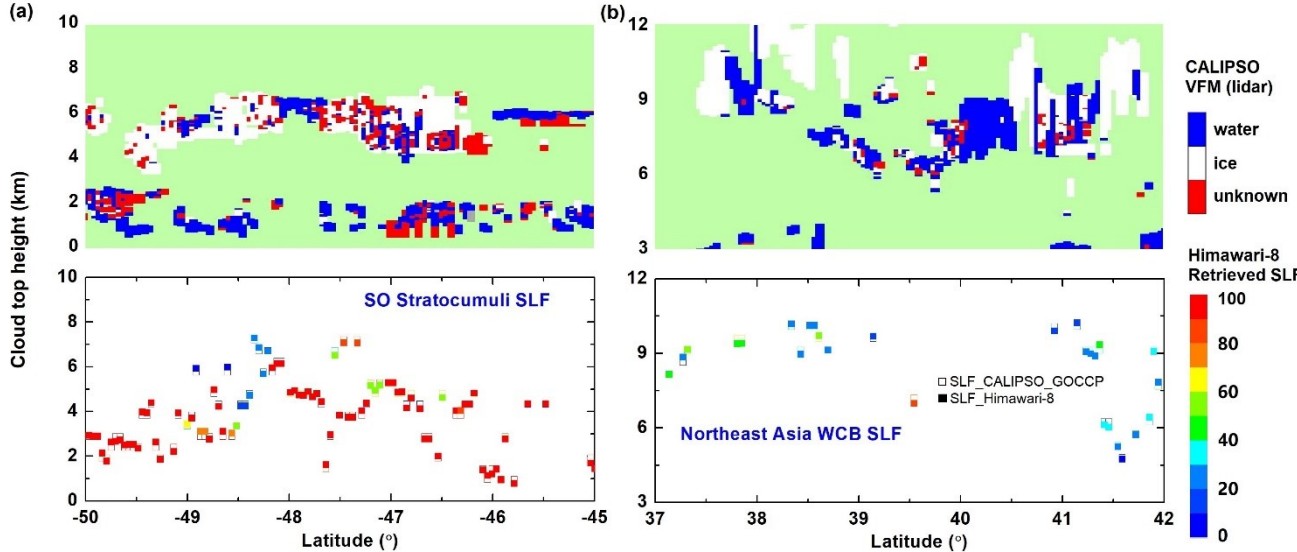

**Figure 6: Cloud phase from CALIPSO VFM, and collocated comparison of the retrieved SLF from Himawari-8 and the CALIPSO-GOCCP product in the vertical section along the CALIPSO overpass on August 28, 2017. (a) SO, 50 ◦S - 45 ◦S. (b) WCB in the mid latitudes of the NH, 37 ◦S - 42 ◦S. The retrieved SLF and the CALIPSO-GOCCP data are represented by solid and hollow squares, respectively. The color of the squares represents the SLF values.**

The question arises: what factors influence the variations in observed values of SLF over the SO and the WCB in Northeast Asia? Possible factors might include thermodynamics, air advection, and background aerosols. To test the feasibility of the method in examining the driven factors under different conditions, we illustrate the seasonal variation in the diurnal cycle of the retrieved SLF in MPCs including the SO's stratocumuli and Northeast Asia cloud systems represented by red and blue rectangles in Fig. 3, which encompass the section of the CALIPSO track at 4:00 UTC (two magenta lines) shown in Fig. 6.





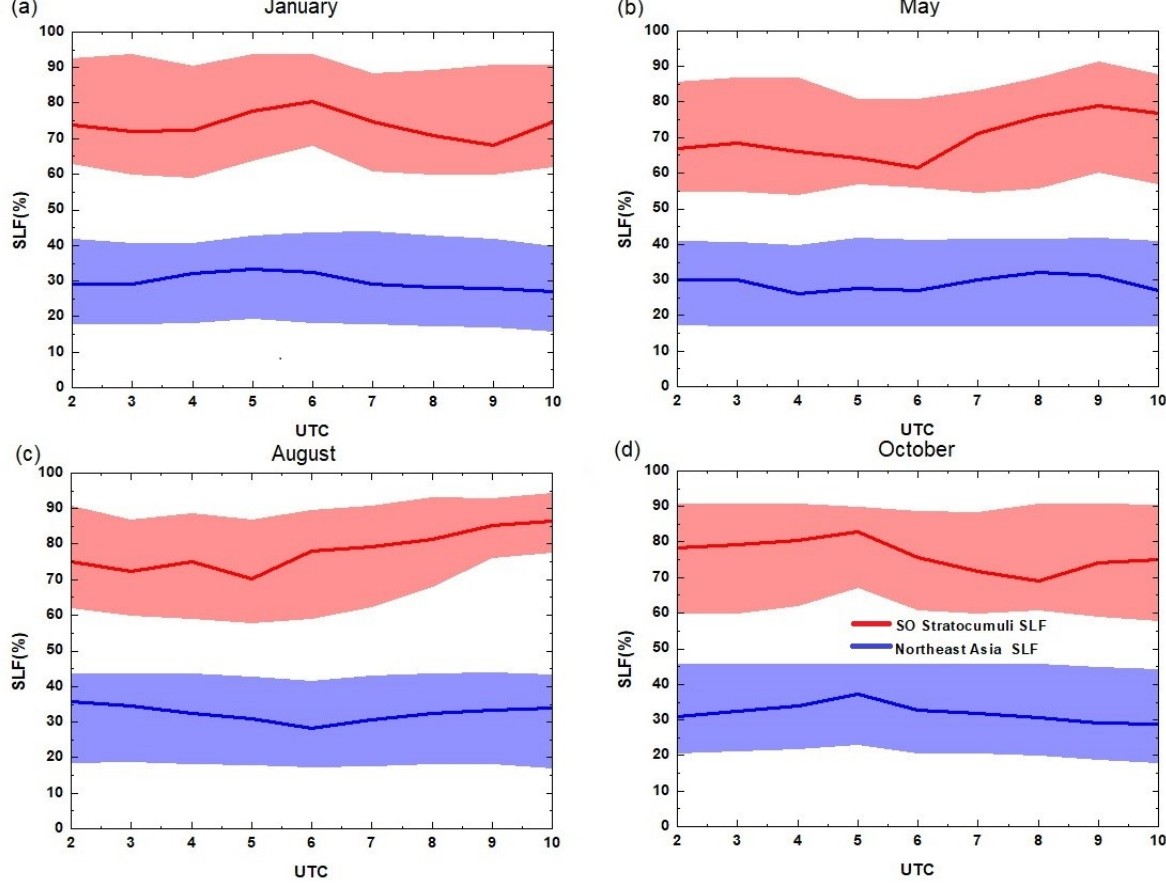

**Figure 7: The diurnal cycle of the SLF (maximum, average, minimum) in MPCs over a larger range in the SO (red pattern) and**
**Northeast Asia (blue pattern) represented by two rectangles in Fig. 3 in (a) January, (b) May, (c) August and (d) October of 2017.**

Initially, we calculate the CTT to identify the temperature bin where the SLF is located. The average CTT in the diurnal cycle
for SO stratocumuli (Northeast Asia cloud regimes) in January, May, August, and October is -9 °C (-19 °C), -24 °C (-14 °C),
-18 °C (-14 °C), and -13 °C (-15 °C), respectively. In the SO, mean CTTs reach a minimum at noon, whereas over Northeast
Asia, the warmest CTTs are observed at 06:00 UTC, which is consistent with the findings in Fig. 11 of Taylor et al. (2017). In
Fig. 7a, the SO's stratocumuli during austral summer display minimal SLF values at early morning and late afternoon, with an
average around 74% and a peak of 81% by 06:00 UTC, which can be explained by the transformation to supercooled liquid
droplets as temperature decreases in the relatively warmer environment. This observation aligns with the results from Noel et
al. (2018) based on measurements obtained from the Cloud-Aerosol Transport System (CATS) lidar aboard the non-sun-
synchronous International Space Station (ISS) to some extent. Noel et al. (2018) indicate that during summer, low-level liquid
cloud droplets at the similar height exhibit a peak occurrence in the early afternoon, approximately around 14:00 local time.
Contrarily, Fig. 7c shows larger SLF at 75% around 02:00 UTC, dropping to a low of 70% until 05:00 UTC, then gradually





rising to 86% by 10:00 UTC in winter. Wang et al. (2022) also use the CATS lidar dataset to analyze the near-global SLF diurnal cycle between 51°S and 51°N at isothermal layers between -10 °C and -30 °C and the minimum and maximum values are consistent with the retrieved SLF here. The vertical motion over the SO is relatively weak, and the atmosphere there

remains stable. Naud et al. (2006) indicated a relationship between the changes in glaciation temperature of supercooled liquid and the surface temperature pattern, and Li et al. (2017) note a positive correlation between SLF and surface temperature at the -10 °C isotherm at middle and high latitudes. This partially explains the increase in SLF values during the austral summer around midday (Fig. 7a), coinciding with warmer surface temperatures. While the decline in SLF during austral winter around noon (Fig. 7c) can be explained by the supercooled liquid transforming into ice around in colder CTT conditions.

Figure 7c displays a decline in SLF from 36% at 02:00 UTC to 28% at 06:00 UTC, subsequently rising to 34% by 10:00 UTC in the NH's summer. In contrast, during austral winter over Northeast Asia in the NH, the SLF peaks at 33% around 05:00 UTC in Fig. 7a. The differences in the diurnal cycle of SLF are complex and can be attributed not only to thermodynamics, but also to dynamics including vertical up and down drafts and advection transport of dust aerosols over Northeast Asia (Li et al., 2017). And Wang et al. (2022) further indicate that the correlation with SLF and meteorological parameters is unstable.

Compared with austral spring, the aerosol effect on nucleation in the SO (Fig. 7b) becomes apparent in austral autumn due to an increase in INPs frequency, such as aerosols from sea spray due to reduced sea ice (Dietel et al., 2023). The SLF value in the SO's stratocumuli generally decreases, ranging between 64% and 79%. In contrast, the SLF in austral spring (Fig. 7d) is generally larger, averaging between 69% and 82%. Over Northeast Asia, the SLF shows reduced values in austral spring (27%-32%, Fig. 7b) compared to autumn (29%-37%, Fig. 7d), likely due to an increase in aerosol frequencies from dust regions

during spring. The trend of the diurnal cycle of SLF in spring and autumn is similar to that in winter and summer, respectively. The average SLF within these two respective cloud systems consistently presents distinct characteristics: notably higher SLF in stratocumulus and lower SLF over Northeast Asia. The SLF in SO stratocumuli in austral spring, summer, autumn, and winter is 76%, 74%, 70%, and 78%, respectively. Over Northeast Asia, the corresponding SLF values are 29%, 32%, 32%, and 30%. The SLF over the SO aligns closely with the values depicted in Fig. 5. However, in the NH, the SLF appears notably

smaller in Fig. 7 compared to the statistical results in Fig. 5, possibly because the selected cloud regimes primarily cover land areas in the NH in Fig. 7, while the statistics in Fig. 5 encompass oceanic regions. Huang et al. (2015) suggest that the difference in the occurrence of supercooled liquid clouds is fundamentally controlled by thermodynamics. Latitudinal and seasonal variations in atmospheric profiles are influenced by dynamic mechanisms (Li et al., 2017; Wang et al., 2022), and the significant influence of aerosols is highlighted in multiple studies (Choi et al., 2010; Hu et al., 2010; Tan et al., 2014; Li et al.,

2017; Villanueva et al., 2020, 2021).

This investigation into the hemispheric and seasonal contrast of the diurnal cycle of SLF in stratocumuli and cloud systems over Northeast Asia is the assessment of the feasibility of the developed method, and would not be feasible without the initial retrieval of SLF using the algorithm first developed in this research for geostationary satellite observations covering a broader





range of spatiotemporal scales than other measurements. The data set can be extended to cover the years 2017 to 2019, allowing

for a comprehensive examination of the observed trend in SLF.

## 4 Conclusions

In this study we present the development of a novel algorithm to characterize the liquid content ratio to the total liquid and ice content, SLF, instead of phase frequency ratio in MPCs using Himawari-8 geostationary satellite observations. The algorithm tests to utilize the different radiative properties of supercooled liquid droplets and droxtal ice particles and applies

them in the process of retrieving supercooled liquid and ice cloud microphysical properties with satellite radiances at VIS and SWI. The CALIPSO-Himawari-8 measurements provide the reliable reference CWP for inferring SLF and IF. With actual retrievals in our case study, the proposed method has been verified, and the SLF can be well constrained. The retrieval results clearly map the distribution of liquid or ice water content in MPCs over representative areas for instance the Southern Ocean. In a broad range of spatiotemporal scales, the cloud phase has a small spatial variability, as observed from satellites, which

presents an optimal scenario for the application of our algorithm. From the zonal comparison results covering different seasons in 2017, our retrieved SLF agrees well with CALIPSO-GOCCP, with the MAE, the RMSE and the correlation coefficient of 9.18%, 10.76%, and 0.69, respectively. The new algorithm shows higher accuracy especially in single-layered cloud systems with a relatively larger deviation of 20% in lower SLF regimes.

To verify the effectiveness of the SLF retrieval algorithm, this study analyzes the seasonal and hemispheric contrast

of the diurnal cycle of SLF in MPCs from distinct cloud regimes. The average SLF of stratocumulus over part of the Southern Ocean ranges between 74% and 78% across various seasons, while that in the cloud systems over Northeast Asia falls within the range of 29% and 32%. In the Southern Ocean, the SLF reaches its minimum around noon in summer and maximum in winter. The trend in the northern hemisphere is different. Previous studies mainly focused on binary cloud thermodynamic phase (Sun and Shine, 1994; Hu et al., 2009; Mouri et al., 2016; Schmit et al., 2017; Mayer et al., 2023) or SLF measurements

and simulations (Shupe et al., 2008; McCoy et al., 2015), or have demonstrated the relationship between SLF changes and meteorological parameters (Li et al., 2017) and investigated the leading importance with respect to cloud climate impact (Tan et al., 2016; Lohmann and Neubauer, 2018). However, investigations into the diurnal cycle of SLF have received far less attention. While some studies, such as the one by Wang et al. (2022), have attempted global-scale assessments of SLF using lidar measurements, tracking the life cycle of MPCs remains challenging. This difficulty stems from the absence of a retrieval

method developed for geostationary satellite observations with the rapid revisit rate required for comprehensive monitoring.

Our results ideally fill this research gap and confirm the feasibility of utilizing passive geostationary satellite remote sensing to retrieve SLF. The previous phase frequency ratio method leads to an upscaling of satellite observations, but our retrieval product can preserve the original resolution. In addition to validating the retrieved SLF along the lidar track, we propose a more comprehensive analysis of the disparities in the global distribution (Himawari-8 disk) of multi-annual maps

(Stengel et al., 2020) between Himawari-8 observations and CALIPSO-GOCCP data. Moreover, augmenting the analysis with



statistical insights derived from longer-term datasets, spanning multiple years, would further consolidate the scientific conclusions regarding the observed trends in SLF. Our approach not only provides the opportunity to track the evolution of MPCs thanks to the high spatiotemporal resolution of geostationary observations but also enhances the ability to constrain the ice production processes during MPCs glaciation in GCMs. This will be the topic for a forthcoming study designed to estimate

cloud climate feedback. Moreover, our method can be improved with the incorporation of a temperature-dependent ice particle habit diagram. Further, our method could be adapted for use with the polar orbiting EarthCARE (Cloud, Aerosol and Radiation Explorer; Wehr et al., 2023) satellite to be launched in 2024, which features both a multispectral imager and an atmospheric lidar instrument on board. This extension opens up new possibilities for global studies in the field of cloud phase climate feedback.


*Data availability*

The Himawari-8 data used for the main SLF retrieval in this study is released from the Japan Aerospace Exploration Agency (JAXA) P-Tree system (JAXA, 2015). The MODIS surface albedo (NASA, 2023), as well as the CALIPSO IIR track data (NASA, 2020a) used for the CWP model building and the CALIPSO Lidar VFM data (NASA, 2020b) used for the analysis

of the cloud phase spatial inhomogeneity are obtained from the Level-1 and Atmosphere Archive & Distribution System Distributed Active Archive Center (LAADS DAAC) via https://ladsweb.modaps.eosdis.nasa.gov/search/ and Atmospheric Science Data Center (ASDC) via https://asdc.larc.nasa.gov/. The CALIPSO-GOCCP "3D_CloudFraction_phase" product used for the validation is derived from Institute Pierre-Simon Laplace (IPSL) (IPSL, 2020). All the data sets are assessed on 3 October 2023. The RSTAR (system for transfer of atmospheric radiation) package is available from the OpenCLASTR project

via http://157.82.240.167/~clastr/. The Random Forest technique is available from https://scikit-learn.org/stable/modules/generated/sklearn.ensemble.RandomForestRegressor.html.

*Author contributions*

Z.W. conceived the study concept. Z.W., S.H., and L.B. developed the methods and designed the experiment. Z.W. conducted

data analyses, visualized results, and wrote the manuscript with suggestions from S.H. and L.B. H.L. contributed to the basic cloud retrieval scheme. S.H. contributed to the Himawari-8 and CALIPSO data collocation. Z.W. and L.B. revised the manuscript based on critical comments from H.L. and S.H. throughout the manuscript.

*Competing interests.* The authors declare that they have no conflict of interest.


*Acknowledgements*

This project has received the funding support by the DLR/DAAD Research Fellowship – Doctoral Studies in Germany, 2020 (57540125) from the joint program of Deutsches Zentrum für Luft- und Raumfahrt (German Aerospace Center) and Deutscher Akademischer Austauschdienst (German Academic Exchange Service), and the Second Tibetan Plateau Scientific Expedition



and Research Program (2019QZKK0206). We acknowledge JAXA and LAADS DAAC for providing Himawari-8/AHI observations, and MODIS surface albedo, CALIPSO IIR and Lidar data, as well as ASDC for supplying the CALIPSO-GOCCP product. We are grateful to OpenCLASTR project for using RSTAR package in this research. We thank Christiane Voigt, Georgios Dekoutsidis, Tina Jurkat-Witschas, and Simon Kirschler from DLR for valuable discussions.

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
