# Peer review of "Technical note: Retrieval of the supercooled liquid fraction in mixedphase clouds from Himawari-8 observations"

_EGUsphere, 2023_

## Author Comment (AC1)

**Responses to anonymous reviewer #1**

We thank all reviewers for their helpful advices and constructive comments about our paper. Their suggestions have led to a revised version of our manuscript where we concentrated on two goals: (1) Supercooled liquid fraction (SLF) in mixed-phase clouds is retrieved for the first time using passive geostationary satellite observations based on differences in liquid droplet and ice particle radiative properties. More information about the algorithm's development and its applicability is provided. (2) The retrieved results are well comparable to global distributions observed by active instruments, and the feasibility of the retrieval method to analyze the observed trends of SLF has been validated. We offer additional quantitative validation results to support this. To this end, we have written concise explanations and modified pictures.

We thank the referee for highlighting the importance of the study and for helpful comments, which we address in the revision of the manuscript.

In the following we enumerate the referee's comments (RC) and reply (R) to them individually.

**RC: Summary of paper:**

This paper presents a novel method for retrieving the supercooled liquid fraction (SLF) of mixed-phase clouds using observations from the geostationary Himawari-8 satellite. By using differences in radiative properties between cloud droplets and ice particles, the method explores the extreme states of clouds, whether fully supercooled or fully glaciated, and reference cloud water content data from CALIPSO are incorporated. The uncertainties of the method are evaluated. Furthermore, the method's accuracy is assessed using various datasets, and its feasibility is thoroughly discussed through applications of analyzing the diurnal cycle of SLF and considering hemispheric contrasts. As a "Technical Note," this paper provides distinctly valuable insights into the further evolution and climate impact of mixed-phase clouds. I recommend its publication in ACP, with some minor comments provided for the authors' consideration, acknowledging the well done work accomplished in this article.

**Comments:**

**RC1:** Introduction: In discussing research on SLF, it is good to include airborne measurements as well, for example, Wang et al. (2023).

**R1a) author's response**

Thanks for pointing out this aspect. We add the suggested research progress on airborne measurements lined to SLF in the introduction, situated between laboratory and field experiments, and global climate modeling. We also update the reference list accordingly.

**R1b)** manuscript changes**

L44-46: "Wang et al. (2023) use post-processed airborne data to characterize the microphysical properties of supercooled liquid droplets and ice particles of stratiform mixed-phase clouds, and indicate that a high concentration of small ice particles can be a result of the secondary ice production in updraft regions."

**References**

Wang, Y., Kong, R., Cai, M., Zhou, Y., Song, C., Liu, S., Li, Q., Chen, H., and Zhao, C.: High small ice concentration in stratiform clouds over Eastern China based on aircraft observations: Habit properties and potential roles of secondary ice production, Atmos. Res., 281, 106495, https://doi.org/10.1016/j.atmosres.2022.106495, 2023.

RC2: Figure 3: Can you imply the relationship between the retrieved SLF and the CWPprediction?

**R2a): author's response**

Thank you for presenting this thought-provoking question. The negative correlation between the retrieved SLF and the CWPprediction has been calculated based on the data provided in Fig. 3. The following plot Fig. A1 displays the relationship between retrieved SLF and CWPprediction with CWPprediction on the x-axis and SLF on the y-axis. Modifications to the text are made in accordance with the given calculations.

**R2b) manuscript changes**

**L236-237: "The values of the retrieved SLF in Fig. 3d generally tend to have a negative correlation with the CWPprediction."**

**RC3:** Line 251-253: "...the cloud phase spatial inhomogeneity is scarcely shown in the Himawari-8-pixel level (5km)". Are you assuming that the clouds within the Himawari-8-pixel level are composed of either liquid droplets or ice particles? This statement may lead to misunderstanding. Could you elaborate further on the implications of cloud phase inhomogeneity in Himawari-8 (5km) from CALIPSO data (333m)?

**R3a): author's response**

Sorry for the unclear description. The MPC clouds at the pixel level of Himawari-8 consist of both liquid droplets and ice particles. However, the used "Feature\_Classification\_Flags" parameter in the CALIPSO Level 2 VFM product only contains the flags for "ice" and "water". The investigation of the cloud phase spatial inhomogeneity is along the CALIPSO orbit track. It is observed that ice and liquid pixels are typically not homogeneously mixed, as seen in the study conducted by Coopman and Tan (2023). Furthermore, the liquid phase scales in the northern hemisphere during May (spring) shown in Fig. 4b, as well as the ice

phase scales in the southern hemisphere during August (winter) shown in Figure 4c, are both notably small. There are inhomogeneities in the Himawari-8 pixel level for these cases, even when using the VFM dataset along the CALIPSO orbit path. To make it clearly the text is now adapted accordingly.

**R3b) manuscript changes**

L260-261: "... and liquid and ice pixels along the CALIPSO orbit track are not homogeneously mixed (Coopman and Tan., 2023) at the Himawari-8-pixel level (5 km)."

**References**

Coopman, Q., and Tan, I.: Characterization of the spatial distribution of the thermodynamic phase within mixed-phase clouds using satellite observations, Geophys. Res. Lett., 50, e2023GL104977, https://doi.org/10.1029/2023GL104977, 2023.

**RC4:** Line 270-274: Would it be better to add a scattering plot or another format of quantitative comparison of "SLF\_CALIPSO\_GOCCP" and "SLF\_Himawari-8" for Figure 6?

**R4a) author's response**

Thanks for commenting on this ambiguity. The comparison between the overlapping pixels of Himawari-8 and CALIPSO-GOCCP here is less intuitive. We have included a quantitative comparison of the data in Fig. 6 for SO stratocumuli and Northeast Asia WCB. The comparison is based on MAE, RMSE, and CC in Table 2. Texts are added accordingly. In addition, we present a scatter plot Fig. A2 (Fig. S4 in the added supplement S3 Comparisons of retrieved SLF and CALIPSO-GOCCP for the selected cases) for two datasets in a same plot as the data amount is small for each case.

**R4b) manuscript changes**

L304-305: "The values of MAE, RMSE, and CC for these two cases are 10.53% and 13.17%, 12.31% and 14.44%, and 0.94 and 0.69, respectively. Please refer to Fig. S4 for details."

Supplement S3: Comparisons of retrieved SLF and CALIPSO-GOCCP for the selected cases

Figure A2 (S4): Comparison of the retrieved SLF from Himawari-8 and the CALIPSO-GOCCP product in the vertical section along the CALIPSO overpass for the selected cases shown in Fig. 6 on August 28, 2017

Technical comments:

RC5: L16: "statistically" instead of "statistical"?

R5a) author's response

Replaced with "statistically" in L16.

**RC6:** L25: Singular verb "forms" should be used here.

R6a) author's response

Replaced with "forms" in L25.

RC7: L141: Plural verb "appear".

R7a) author's response

Replaced with "appear" in L144.

RC8: L146: "analyzed" instead of "analyze".

R8a) author's response

Replaced with "analyzed" in L149.

RC9: L202: "varies" instead of "vary".

R9a) author's response

Replaced with "varies" in L204.

RC10: L316: with an average "of" around 74%...

R10a) author's response

Added "of" in L326.

RC11: L329: remove "around".

R11a) author's response

Removed "around" in L339.

**Responses to anonymous reviewer #2**

We thank all reviewers for their helpful advices and constructive comments about our paper. Their suggestions have led to a revised version of our manuscript where we concentrated on two goals: (1) Supercooled liquid fraction (SLF) in mixed-phase clouds is retrieved for the first time using passive geostationary satellite observations based on differences in liquid droplet and ice particle radiative properties. More information about the algorithm's development and its applicability is provided. (2) The retrieved results are well comparable to global distributions observed by active instruments, and the feasibility of the retrieval method to analyze the observed trends of SLF has been validated. We offer additional quantitative validation results to support this. To this end, we have written concise explanations and modified pictures.

We thank the reviewer for his/her positive judgement on the manuscript and the helpful comments.

In the following we number the referee's comments (RC) and reply (R) to them individually.

**RC: Summary of paper:**

The authors present a method for obtaining the mass phase ratio (MPR) of clouds in the mixed phase temperature regime from the Himawari-8 satellite. They compare this method with CALIPSO-GOCCP and show good first results. The method is novel and timely, and this technical note is worthy of publication. However, a weakness of the evaluation is that there are no comparisons between SLF and temperature, which makes it difficult to assess the quality of the results. Nevertheless, as a technical note focusing on the novelty of the method, I support the publication of the paper in ACP.

**Ra) author's response**

Thank you for emphasizing the originality of this approach and acknowledging the flaw in not considering the correlation between SLF and temperature. We employ the dataset of retrieved SLF shown in Fig. 5, along with the cloud top temperature (CTT) data from Himawari-8, to assess the dependence of SLF on temperature. The related plot Fig. A3b is integrated with Fig. S1 as Fig. S1b in the supplement. The CTT-dependent SLF retrieved at the pixel-level using our method aligns the findings derived from adjacent pixels in Choi et al. (2010) and Han et al. (2023). We plan to have more detailed analysis in the future.

Figure A3 (S1): CTT-dependence of cloud phase. (a) Densities of Himawari-8 cloud phase as a function of CTT. Blue and red lines mark the temperatures 233 K and 273 K. (b) CTT-dependence of retrieved SLF values.

**Rb) manuscript changes**

L15-21 in the supplement: "Fig. S1b shows the CTT-dependence of SLF in the detected MPCs observed by Himawari-8. The SLF values are approximately 100% above -10 °C and decrease towards 0 as temperatures drop below approximately -30°C. Within the temperature range of -35°C to -15°C, SLF shows an increasing trend. Notably, the retrieved SLF exhibits a small minimum between -10 and -3 °C that could be related to rime splintering in secondary ice production. The observed SLF trend as a function of CTT closely resembles findings in Choi et al. (2010) and Han et al. (2023). Pixels exhibiting uncertainties for retrieved negative values, as mentioned in Sect. 2.3, have been filtered out before conducting this detailed investigation."

**References**

Choi, Y. S., Lindzen, R. S., Ho, C. H., and Kim, J.: Space observations of cold-cloud phase change, P. Natl. Acad. Sci. USA, 107, 11211–11216, https://doi.org/10.1073/pnas.1006241107, 2010.

Han, C., Hoose, C., Stengel, M., Coopman, Q., and Barrett, A.: Sensitivity of cloud-phase distribution to cloud microphysics and thermodynamics in simulated deep convective clouds and SEVIRI retrievals, Atmos. Chem. Phys., 23, 14077–14095, https://doi.org/10.5194/acp-23-14077-2023, 2023.

**Some minor suggestions:**

**RC1:** Line 108: This would be clearer if the sentence were split into two sentences explaining input and output.

**R1a) author's response**

We realize that the long sentence in this context causes confusion for the reader. We adhere to the recommendations and divide the content into two separate sentences, one focusing on input and the other on output. Changes in the text are given as follows.

**R1b) manuscript changes**

L113-117: To build the CWP prediction model for the full disk of Himawari-8, "the input consists of AHI channel and geometrics data, including solar zenith angle (SZA), viewing zenith angle (VZA), and relative azimuth angle (RAA), as well as Aqua MODIS 16-day averaged surface albedo with a resolution of  $0.05^\circ$ , which is adjusted to match the AHI resolution using the closest neighbor approach. The output of the model is represented by the variable  $CWP_{CALIPSO}$ . Table 1 provides specific information regarding the data."

**RC2:** Line 130: Please check the trailing "IF" at the end.

**R2a) author's response**

Thank you for bringing up this vague definition of abbreviation. We initially define the term "ice fraction" as "IF" at line 78 in the introduction. We now establish the abbreviation in the dataset description in Sect.2.1 Data collocation and preprocessing. The text is modified accordingly.

**R2b) manuscript changes**

L78: Remove "(IF)".

L133: Replace "IF" as ice fraction "(IF)".

RC3: Line 209: Ice-dominant -> Ice-dominated. belt -> belts

**R3a) author's response**

We have implemented the corrections suggested by the reviewer, replacing "ice-dominant" with "ice-dominated" and "belt" with "belts", respectively, in L212.

**RC4:** Line 210: Which invalid cases are excluded? Phase-overlap scenes, ice-dominant MPCs or WCBs and convection? How are these cases defined? Please clarify

**R4a) author's response**

Thank you for inquiring about clarification. Cases exhibiting negative values of SLF are consequently excluded from the statistical analysis, as stated in the current version of manuscript. In these cases, the ice fraction computed by the algorithm exceeds 100%. To investigate the reason for these unphysical values, we utilized all available Himawari-8 data in this study and computed the ratio of phase-overlap scenes among all invalid cases. The ratio of phase-overlap in all invalid retrievals is calculated to be 0.63.

Additionally, detecting the liquid fraction in ice-dominated MPCs presents challenges. Providing systematic detection results for WCBs and convection within ice-dominated MPCs is not that easy. Instead, Fig. 3 on August 28, 2017, serves as a visual reference for interpretation. Invalid retrievals, identified by blue regions with values approaching 0, are indicative of WCBs over Northeast Asia.

**R4b) manuscript changes**

L213-218: "In all invalid cases, the retrieved SLF manifests as negative values, resulting from IF exceeding 100%. Phase-overlap contributes to 63% of the total number of these cases. Focusing on ice-dominated MPCs, including WCBs and convection, we utilize Fig. 3 in the subsequent section as a visual demonstration. The corresponding retrieved SLF, approximately 0% in Fig. 3d, aids in interpreting the uncertainties of retrieval in ice-dominated MPCs within the Northeast Asia region depicted in Fig. 3b." Negative values of SLF are set to zero for these invalid "retrievals in phase-overlap scenes and ice-dominant MPCs", and they are consequently excluded from the statistical analysis.